# Scalable Learning Framework for Detecting New Types of Twitter Spam with Misuse and Anomaly Detection

**DOI:** 10.3390/s24072263

**Published:** 2024-04-02

**Authors:** Jaeun Choi, Byunghwan Jeon, Chunmi Jeon

**Affiliations:** 1College of Business, Kwangwoon University, Seoul 01897, Republic of Korea; juchoi@kw.ac.kr; 2Division of Computer Engineering, Hankuk University of Foreign Studies, Yongin 17035, Republic of Korea; 3Corporate Relations Office, Korea Telecom, Seoul 03155, Republic of Korea

**Keywords:** Twitter spam, anomaly detection, decision tree, autoencoder

## Abstract

The growing popularity of social media has engendered the social problem of spam proliferation through this medium. New spam types that evade existing spam detection systems are being developed continually, necessitating corresponding countermeasures. This study proposes an anomaly detection-based framework to detect new Twitter spam, which works by modeling the characteristics of non-spam tweets and using anomaly detection to classify tweets deviating from this model as anomalies. However, because modeling varied non-spam tweets is challenging, the technique’s spam detection and false positive (FP) rates are low and high, respectively. To overcome this shortcoming, anomaly detection is performed on known spam tweets pre-detected using a trained decision tree while modeling normal tweets. A one-class support vector machine and an autoencoder with high detection rates are used for anomaly detection. The proposed framework exhibits superior detection rates for unknown spam compared to conventional techniques, while maintaining equivalent or improved detection and FP rates for known spam. Furthermore, the framework can be adapted to changes in spam conditions by adjusting the costs of detection errors.

## 1. Introduction

Owing to recent advances in wired and wireless communication technologies and the global popularity of smartphones, the number of wired and wireless online users has reached five billion, i.e., 63% of the world’s population, as of April 2022. Consequently, the number of social media users communicating online on platforms such as Facebook, Twitter, Instagram, Snapchat, and WeChat has also increased substantially to 4.65 billion people, comprising 93% of global Internet users [1]. Social media users communicate via posts, chats, and comments and share their feelings through likes, dislikes, tags, and follows. Furthermore, social media services are used for professional activities, such as sharing professional links, exchanging opinions, increasing business reputation, running promotions, online learning, sharing news, managing disasters, and operating expert systems [2].

With the increase in the volume of daily and professional activities performed on social media, cyberattacks targeting these activities have also been on the rise. Spammers often publish simple illegal advertisements on social media [3], and automated bots have emerged that create social problems by spreading fake news, aggravating political turmoil, disrupting stock markets, and enabling cyberbullying [4,5]. Several accounts have been reported for spreading political propaganda and disinformation during elections [6]. In one incident, approximately 130 high-profile accounts, including those of Barack Obama, Joe Biden, and Elon Musk, were hijacked and used to tempt victims to make fake Bitcoin transactions [7]. Incorrect and unverified medical information was also spread quickly through social media during the COVID-19 pandemic, creating confusion and unnecessary fear [2].

In the dynamic landscape of social media, spam distributors are adept at innovating new types of spam to circumvent existing defense strategies. A notable manifestation of this adaptability is the prompt exploitation of emerging online trends to fabricate novel spam forms. For instance, amidst the COVID-19 pandemic, a proliferation of related spam was observed, swiftly capitalizing on the global crisis [2]. Similarly, the ascendancy of cryptocurrencies, such as Bitcoin, saw a corresponding increase in spam exploiting this phenomenon [7]. More recently, the advent of generative artificial intelligence technologies, including platforms like ChatGPT, has marked the emergence of a new vector for spam creation [8]. This ongoing evolution presents a significant challenge for spam detection, as newly developed spam, characterized by distinct and previously unseen features, evades detection by traditional methodologies. This study proposes an effective method to detect new and unknown types of spam on social media, with a particular focus on Twitter, where the prevalence of spam is widespread. Notably, although Twitter was acquired by Elon Musk and subsequently rebranded to X in July 2023, this study references data collected from the platform prior to this change; thus, we refer to it as ‘Twitter’ throughout this manuscript for consistency and clarity.

Various methods have been proposed to detect spam on social media sites such as Twitter. Traditional methods include the blocklist method, which blocks malicious URLs or internet protocol addresses. However, creating a blocklist is both time- and effort-intensive, and with the widespread use of short/tiny URLs in recent years, attacks that evade blocklists have become more frequent [2,9]. Alternatively, spam accounts may be detected by installing a honeypot at an appropriate location on social media [10,11]. A honeypot is a system deliberately installed to lure attackers and is widely used for attack and spam detection in networks. However, honeypot-based methods suffer from several issues, such as poor adaptability and scaling, high volume of data processing, and poor portability. Their implementation is also time-consuming as they require expert intervention [2,12]. To overcome such drawbacks of traditional methods, recent studies have employed and developed machine learning (ML) and deep learning (DL)-based methods to detect spam and malicious behavior on social media.

ML and DL-based spam detection methods can be divided into spam content-based and spammer account-based detection methods. Spam detection techniques based on extracting information from both content and accounts have also been proposed [2,13]. These techniques employ various algorithms, ranging from traditional ML methods to the latest DL methods based on extracted features, and most exhibit excellent performance. Most studies utilizing ML and DL employ misuse detection, which identifies behaviors matching those of known attacks. When misuse detection is used, excellent performance is achieved in detecting known attacks because previously known data are learned and subsequently used for spam detection. However, misuse detection methods are ineffective against previously not learned spam attacks evolving in real time, and their major disadvantage is performance degradation due to class imbalance caused by imbalanced amounts of spam and non-spam data [12,14]. In particular, spammers have collaborated and rapidly shifted their attack strategies in response to recent defense strategies. To respond to rapidly evolving spamming, there is a demand for anomaly detection utilizing ML and DL [2]. As anomaly detection algorithms identify behaviors deviating from those corresponding to normal data, they can detect unknown data that are different from normal data. However, they exhibit poor performance when used in isolation because they only learn normal data patterns [2].

This paper proposes a hybrid method that uses anomaly detection and misuse detection to respond to new spam attacks. Anomaly detection makes decisions after learning unlabeled data. ML and DL-based anomaly detection is widely used in various tasks, such as fraud, network anomaly, and intrusion detection [15]. We use a one-class support vector machine (SVM) and an autoencoder, which are typical anomaly detection methods, to detect new spam. In anomaly detection, normal data are learned, and data with different properties are classified as abnormal. This approach can be leveraged to detect unknown spam types. However, its detection rate is low because it creates a single model using a broad range of normal data. In this paper, the detection rate denotes the ratio of correctly classified spam to the overall actual spam data. Furthermore, its false positive (FP) rate, which is the probability of classifying a normal data point with abnormal properties as abnormal, may be high. A high FP rate degrades a system’s usability from the user’s perspective. Thus, achieving a high detection rate while keeping the FP rate low is important for efficiency and reliability. Besides improving user trust in the system, it also ensures that valuable communication is not hindered. To balance these two metrics, this study first employs a decision tree (DT), an ML algorithm widely used for misuse detection, to improve these poor performance metrics. A DT divides data into leaf nodes by splitting branches. The training samples are mapped to leaf nodes until classification is complete, and the validation data are transmitted to the leaf nodes according to the DT branch generation criteria created during training to produce the classification results.

We classify the data into known and unknown spam using a DT trained on known spam and normal data. Data not corresponding to known spam are subsequently classified as unknown spam or normal tweets using anomaly detection trained on normal data. By first detecting known attacks using DTs, we adequately compensate for the poor spam detection rate of anomaly detection. Data different from known spam can be grouped into decomposed subsets using DTs. Moreover, as anomaly detection is applied to each subset, its detection rate for new attacks is increased while reducing its FP rate. During the detection of known attacks primarily using DTs, the proposed framework can adjust the attention of the DT by adjusting the costs based on the current spam situation. For example, if new attacks do not occur frequently and detecting known attacks is essential, the cost of incorrectly detecting known spam can be set to be high in the DT, enabling the detection of as many known attacks as possible. Conversely, if new spam is being produced continuously, the ambiguous data in the DT can be subsequently processed via anomaly detection to detect new attacks. Using this adjustable hybrid method, we propose a system that can respond appropriately to situations requiring both misuse and anomaly detection.

The main contributions of this article are as follows:(1)We propose a spam detection framework based on an autoencoder and a one-class SVM, which are typical anomaly detection methods, to respond to new types of spam that pose a significant threat.(2)We use a DT to respond to known spam and enhance the low spam detection rate of anomaly detection.(3)We increase the detection rate of known spam and normal tweets by performing tailored anomaly detection for each subset containing data not classified as spam by the DT.(4)We propose a scalable spam detection framework that focuses on known or unknown spam, depending on the current situation.

The remainder of this paper is organized as follows. Previous studies on misuse and anomaly detection methods for spam are reviewed in Section 2 and the necessity of the proposed hybrid method is discussed. The proposed Twitter spam detection method is introduced in Section 3. Finally, the experimental results are presented in Section 4 and our conclusions as well as directions of future research are discussed in Section 5.

## 2. Literature Review

### 2.1. Misuse Detection

Numerous ML methods have been proposed for detecting social media spam and this remains an active topic of research [2]. Many ML-based spam detection studies have been proposed based on misuse detection. Traditional misuse detection methods utilize DTs, naïve Bayes methods, SVMs, and random forests (RFs). DL methods actively studied in recent years, such as convolutional neural networks (CNNs), recurrent neural networks (RNNs), and deep neural networks (DNNs), have also been widely used. ML and DL algorithms for misuse detection extract features from labeled data that have already been classified as spam or normal data. Owing to the diversity of features extracted from social media, many studies have applied ML and DL-based techniques after extracting appropriate feature sets. Most feature sets include features obtained from Twitter content; information can also be extracted from Twitter accounts. Recently, some studies have analyzed relationships among accounts using social graphs as well as message contents using text mining and used the results as features.

The most fundamental feature of ML-based methods is the statistical information obtained from tweets. Statistical information is considered because typical spam employs a social engineering technique that transforms the text narratives of tweets [13]. Therefore, spam detection is possible by using statistical characteristics as features. For example, spam tweets tend to include multiple hashtags and URLs or contain the word “spam”. Thus, these characteristics can be used as features [16]. Chen et al. analyzed six million tweets and suggested the following features: number of tweets posted, number of retweets, number of hashtags, number of user mentions, number of URLs in the tweet, proportion occupied by URLs in the tweet, and number of letters and numbers contained in the tweet [17]. Multiple spam-detection studies have analyzed features extracted based on hashtags, extractable statistical information, and basic text information, although different studies have used slightly different features [18,19]. Some studies have considered the statistical information within the text as well as the text itself, using text-mining methods such as word2vec or long short-term memory to detect spam [20,21]. Further, a method based on text mining was proposed to prevent illegal URLs from being disseminated via short-message services [22]. Recently, a spam detection technique utilizing large language models using contextualized embeddings such as BERT and ELMo has also been proposed [23]. Features obtained from tweets have been used for text spam detection in various studies because they intuitively reflect spam characteristics and are relatively easy to extract.

Several other studies have focused on spammers posting spam tweets. Spammers often exhibit characteristics different from normal users, e.g., they usually have recently created accounts and few followers. Wu et al. proposed a spammer detection method based on the account’s age, number of followers, followed accounts, and rating [12]. Because an account’s statistical information, which intuitively reflects characteristics that signal it as a spam account, can be extracted easily, some studies have used this in conjunction with the statistical information of an account’s tweets (as described above) as the feature set [24,25,26].

Spam detection studies have also been conducted based on combinations of the aforementioned features or completely novel features, with the number of DL-based studies on the rise. A spam detection method that uses an ensemble of CNNs after extracting account information, tweet information, and tweet n-grams as features was proposed in [27]. Jain et al. proposed a method for detecting Twitter spam using a semantic CNN and word2vec [28]. Another study detected social bots using DNNs and active learning based on various features, such as metadata, interaction, and content [29]. Some studies have used CNNs to detect aggressive behavior on social media [30], and DL-based methods, such as CNNs, have also been proposed to detect malicious images, such as deepfakes [31,32]. Meanwhile, some studies have suggested social graph-based methods that detect spam accounts by analyzing inter-account relationships [33,34,35]. Recently, a spam detection technique using Transformer has also been proposed based on the multi-head attention mechanism [36].

Although misuse detection methods use different features and techniques, they respond well to known attacks. However, as these methods rely solely on learning existing attack patterns, they may struggle to detect variations, evolved versions, or combinations of known attacks. In particular, ML-based misuse detection methods typically struggle with respect to newly emerging attacks [37]. Spam detection methods using anomaly detection have been proposed to overcome this shortcoming.

### 2.2. Anomaly Detection

Anomaly detection methods are widely used for identifying abnormal data in various security tasks, such as spam detection. Anomaly detection detects new attacks by identifying abnormal samples after learning the features of normal data [38]. Traditionally, clustering algorithms have been used for anomaly detection. In recent years, DL-based autoencoders have also been used for this task in various fields.

Sohrabi and Karimi used various clustering techniques to detect spam in Facebook comments [39]. Another study proposed the Twitter spam detection framework, INB-DenStream, which outperforms conventional clustering techniques, such as CluStream, DenStream, and StreamKM++ [40]. Research has also been conducted to detect obfuscated words in spam messages using the rule-based hash forest algorithm and rule encoding techniques [41]. Currently, misuse detection-based spam identification methods outnumber anomaly detection-based ones significantly, as the latter ones do not perform well in isolation. In particular, clustering algorithms, which are the most popular algorithms for anomaly detection, exhibit low detection rates and high FP rates when used in isolation [2]. Some studies have proposed hybrid methods based on learning both labeled and unlabeled data to overcome this disadvantage.

Singh and Batra proposed a spam detection method based on probabilistic data structures, reducing the number of computations using filters based on URL and hashtag information [42]. Yilmaz and Durahim proposed a solution utilizing semi-supervised learning. This solution can determine spam reviews based on features extracted from the review text and reviewer product network [43]. A hybrid system for anomaly detection was also proposed to identify abnormal users on social networks [44]. Compared to traditional ML-based solutions, these solutions exhibit greater applicability as they also learn unlabeled data. However, as known attack data are used for detection, these methods still struggle to detect new types of attacks.

To detect new spam, adjustments to the data learning cycle have also been proposed. In the spam detection method proposed by Sedhai and Sun, a module that updates the model was constructed to respond to new spam [45]. Further to this, some studies have proposed periodically updating the learning model to respond to new types of spam [12,25]. Because these methods include spam detection and update modules, they can learn new types of spam robustly; however, their vulnerabilities can be exposed depending on the update cycle. Responding to such vulnerabilities can be difficult when new spam appears faster than the update cycle. In order to compensate for the shortcomings of anomaly detection when used in isolation, this study proposes a hybrid method that uses anomaly detection to detect new spam and combines it with misuse detection to improve performance.

## 3. Materials and Methods

### 3.1. Model Design Overview

This study proposes a framework that utilizes both anomaly and misuse detection to detect new spam effectively, exhibiting a high detection rate and low FP rate. As illustrated in Figure 1, the proposed framework consists of three modules: (i) a known spam detection module based on DT, (ii) an unknown spam detection module based on anomaly detection, and (iii) a scalable spam detection system. In this paper, ‘known spam’ refers to collected tweets already labeled as spam and ‘unknown spam’ refers to spam tweets with characteristics different from those of normal tweets as well as previously known spam.

First, a DT is used to detect known spam effectively. During Stage 1, known spam is filtered out with a high detection rate, and the unfiltered data are grouped into multiple subsets with similar properties. An anomaly detection algorithm trained only on a normal model is applied to each subset. This enables normal and unknown spam data to be distinguished. In this study, in addition to the widely used one-class SVM, we use an autoencoder for anomaly detection to improve the detection rate for new spam. Furthermore, the proposed spam detection system can be adjusted based on the requirements by adjusting the cost arising from the classification error in the DT during the first stage. The detection of known or unknown spam can be facilitated by adjusting the DT target accordingly. The cost-based scalable system is implemented in Stage 1; however, we categorize it as a separate module to explain its theory and usefulness in detail. The modules are discussed in detail in the following subsections.

### 3.2. Known Spam Detection Using DT

During the first stage, the proposed framework detects known spam effectively. We use a robust algorithm, e.g., DT, for classification, regression, and multi-output tasks following training using a complex dataset [46]. It is crucial to ensure that known spam is detected by the framework’s DT and that the remaining data are transmitted to Stage 2.

During Stage 1, our framework first creates a tree structure by learning known spam and normal data. Subsequently, incoming tweets are verified using this tree. Spam detection is considered to be successful if the incoming tweet is classified as known spam by the DT. If an input is not classified as known spam, it can be classified as a normal or unknown spam tweet via anomaly detection in the second stage.

As depicted in Figure 1, the DT begins with the tree’s root, creates child nodes via branches, and continues branching out to the leaf nodes that make predictions. The final prediction is determined at a leaf node at the end of the tree, which has no further child nodes. While branching out, the threshold values of the features are selected at their respective nodes. The data are transmitted to the lower-left or -right nodes, depending on whether the feature’s threshold is exceeded or not.

Typical DT training algorithms include Gini impurity-based classification and regression tree (CART) [47] and entropy-based C4.5 [48]. These methods create similar trees; CART performs calculations slightly faster than C4.5, which requires logarithmic computations [46]. In this study, we implement a CART-based DT. The CART algorithm divides the training dataset into two subsets using the threshold value, $t_{k}$, of each feature, *k*. The cost function that must be minimized to identify the pair, (*k*, $t_{k}$) is as follows:(1)$$J\left(k,t_{k}\right)=\frac{m_{left}}{m}G_{left}+\frac{m_{right}}{m}G_{right},$$
where *J* denotes the cost function based on the values of *k* and $t_{k}$, $G_{left/right}$ denotes the Gini impurity of the left/right subset, and $m_{left/right}$ denotes the number of samples in the left/right subset.

The Gini impurity is used to determine the DT branches. The goal is to identify the split with the lowest impurity value. If the Gini impurity is 0, the node consists of only one class. Values of the Gini impurity closer to zero are considered to be better. The Gini impurity of the *i*-th node is calculated using the following equation:(2)$$G_{i}=1-\sum_{k=1}^{n}{P_{i,k}}^{2},$$
where $p_{i,k}$ denotes the percentage of samples belonging to class *k* of the training samples at the *i*-th node. The DT used in our framework classifies spam into known and unknown spam classes, corresponding to *k* = 1 or 2, respectively. CART determines the final tree structure by repeatedly dividing the training dataset using the aforementioned steps.

CART is trained to distinguish known spam from other tweets during Stage 1. The known spam tweets detected in Stage 1 and the remaining data transmitted to Stage 2 are classified during the test process. To this end, we construct a scalable system using cost-based DTs, which can be specialized to focus on either existing or new spam, depending on the current situation. This functionality is explained in detail in Section 3.4. One advantage of this approach is that known spam can be detected using CART in Stage 1. Another advantage is that a DT is a white-box model, where the decision-making method is Intuitive and easily comprehensible [46]. As DTs perform well and the computation of the predictions is verifiable, the data transmitted to Stage 2 can be effortlessly grouped into multiple subsets with similar characteristics. Furthermore, this improves the detection rate during Stage 2 because anomaly detection is performed for each subgroup. In Section 3.3, we explain this improvement in the detection rate.

### 3.3. Unknown Spam Detection Using Anomaly Detection

The DT detects known spam in Stage 1, and the remaining data are transmitted to Stage 2. In Stage 2, anomaly detection is performed to classify normal tweets and unknown spam types. In the learning process, subsets of known spam and subsets of normal tweets are divided at the terminal leaf nodes of the DT. Anomaly detection is applied to each subset of normal tweets to learn the normal patterns. In the subsequent test stage, the data transmitted to Stage 2 are verified using the normal model corresponding to the leaf node where they are located to classify them as normal data or unknown spam.

As described earlier, when anomaly detection is used in isolation, it exhibits poor detection rate and high FP rate. An examination of the actual data reveals that the patterns of normal tweets are also highly diverse. If these diverse normal tweets are modeled using one model, the model is trained to be sensitive to normal tweets’ characteristics, increasing the FP rate. Conversely, the spam detection rate decreases if the normal model is smoothed excessively to reduce the FP rate. To circumvent these problems, similar normal data are classified into separate subsets using a DT in this framework. Subsequently, the normal data of each subset are modeled. Normal data are learned by concentrating on each decomposed region and subsequently used for detection, thereby increasing detection performance. This improvement is illustrated in Figure 2, with (i) an illustration of learning the entire normal dataset and using it for detection and (ii) an illustration of learning and modeling the normal data of the subsets of each decomposed region and using them for detection. This framework exhibits high accuracy because it models normal data more precisely.

We use a one-class SVM and an autoencoder for anomaly detection. The one-class SVM is a variation of the SVM method known for its high performance and was developed for anomaly detection. It creates a hyperplane enveloping the region of data with one class. This method is selected owing to its suitability for outlier detection [46]. In addition to the traditional method, a DL-based autoencoder, which has been recently implemented as an anomaly detection method, is also used. The proposed framework can detect known and unknown spam effectively using a DT and a one-class SVM or using a DT and an autoencoder, respectively. The two anomaly detection methods are described in the following subsections.

#### 3.3.1. One-Class SVM

One-class SVMs are widely used in detecting structural damage [49], faults [50], and network intrusion [51]. They are derived from typical SVM classifiers and were proposed by Schölkopf et al. [52]. When a typical kernel SVM classifier separates two classes, it maps all samples to a high-dimensional space. Then, it identifies a hyperplane that separates the two classes in the high-dimensional space using the SVM classifier. The one-class SVM algorithm separates the samples in the high-dimensional space from those in the original space because only one class of samples exists. It then identifies a hyperplane enveloping all the samples. If a new sample does not lie within this region, it is deemed to be an abnormal data point [46].

Let us take a closer look at the process of identifying a suitable hyperplane. Let $x_{i}$ denote an item in the training data; χ, the original space; and l, the number of elements in the training data. The SVM uses the feature map (Φ: χ → F) to transform the original high-dimensional space into a feature space non-linearly. The one-class SVM is formulated using the following quadratic program:$$\underset{w,\xi ,\rho }{\min}\frac{1}{2}{\left\|w\right\|}^{2}+\frac{1}{vl}\sum_{i=1}^{l}\xi_{i}-\rho$$
(3)$$subject\;to\left(w\cdot \Phi \left(x_{i}\right)\right)\geq \rho -\xi_{i},$$
$$\xi_{i}\geq 0,i=1,\cdots ,l$$
where w denotes the vector orthogonal to the hyperplane, ξ denotes the vector’s slack variable, and *ρ* denotes the margin from the origin to the hyperplane. The parameter v denotes the proportion of rejected training data and $\left(1-v\right)\times 100\;\%$ represents the training data included within the hyperplane. This framework creates a hyperplane of normal data for each subset of normal data gathered during the training process. When incoming tweets lie outside the normal hyperplane category, they are considered to be outliers and classified as unknown tweets.

Because the one-class SVM divides the decision boundary non-linearly, it works well even with high-dimensional datasets, such as spam data. However, the FP rate is high if all the data are represented within a single hyperplane with various normal data types. The proposed method creates subsets of similar normal tweets using DT. Hence, each normal tweet subset’s model is less complex than that for all normal tweets. Because training and discrimination are conducted using the one-class SVM for each subset, the FP rate is reduced and the performance of anomaly detection is increased.

#### 3.3.2. Autoencoder

An autoencoder is a deep learning algorithm frequently used for anomaly detection. Unlike traditional ML methods, an autoencoder reconstructs an input to be as close as possible to the original rather than extracting its class or target value [53]. The autoencoder achieves this by performing an encoding process that compresses the input to a code or a latent variable, and subsequently a decoding process that yields a maximally similar reconstruction. This process enables essential information about the data to be learned in a compressed manner. The differences between the input and output data are compared using encoding and decoding. Multiple hidden layers can be used during the encoding and decoding processes, as illustrated in Figure 3 [54].

Because the autoencoder restores data to be as close to the original as possible, it learns the features of the normal region, which is the principal region related to the data. If a normal data sample is entered as the input into the trained autoencoder, the input and output are almost indistinguishable. However, when an abnormal data sample is entered, the autoencoder attempts to reconstruct it as if it were a normal sample, resulting in certain differences between the input and output. Thus, if the difference between the input and output exceeds a certain threshold, the input can be considered to be abnormal data. Because autoencoders do not require separate labeling and can automatically distinguish between normal and abnormal data, they are widely used for anomaly detection in various fields [55,56,57,58,59]. This study leverages these autoencoder properties to detect unknown spam tweets.

The autoencoder comprises multiple hidden layers; its *L* layers can be divided into two parts. The first and second half of the layers perform the encoding and decoding roles, respectively. The output, *z*, of the *l*-th layer located in the autoencoder can be calculated as follows [53]:(4)$$z^{l}=f\left(W^{l}z^{l-1}+b^{l}\right),$$
where $W^{l}$ denotes the connecting layer’s weight matrix, $z^{l-1}$ denotes the (*l* − 1) layer’s output, and $b^{l}$ denotes the *l*-th layer’s bias vector. Moreover, f(x) denotes an activation function, which is often taken to be the sigmoid or softplus function [53,60].

The autoencoder is used to determine the difference between the input and reconstructed values, allowing us to determine whether a data sample is normal or abnormal. The outlier factor ${OF}_{i}$ is proposed to check whether a data sample is an outlier [61]. ${OF}_{i}$ is defined to be the average reconstruction error for all features of the *i*-th data sample as follows:(5)$${OF}_{i}=\frac{1}{n}\sum_{j=1}^{n}{\left(x_{ij}-o_{ij}\right)}^{2},$$
where *x* and *o* represent the input and output, respectively, and *n* represents the total number of features. If the extracted outlier factor exceeds a certain threshold, the data sample is considered to be an outlier, i.e., a new type of abnormal data. Various methods have been proposed to determine the threshold, ranging from methods using the mean (*μ*) and standard deviation, median (med) and standard deviation, and first (*Q*1) and third (*Q*3) quartiles [60]. Since *Q*3 − *Q*1 describes a dataset’s degree of dispersion, μ+1.5(Q3−Q1) and med+1.5(Q3−Q1) can be used as thresholds.

In the autoencoder, multiple layers are often used for anomaly detection because this approach handles high-dimensional data effectively [57]. This study uses a multilayer-based autoencoder to increase the detection rate of unknown spam in the second stage, as illustrated in Figure 1. Furthermore, the detection performance is improved because DT is first used to divide normal tweets into multiple subsets according to their characteristics, and anomaly detection is subsequently performed using the autoencoder for each subset.

### 3.4. Scalable Spam Detection System

In the first stage, the proposed framework detects known spam using DT. Anomaly detection is performed on the remaining data, separating them into normal and new spam tweets. In most spam-detection studies, classification is performed during the ML stage, after which the system is terminated. In contrast, our framework detects known spam tweets in Stage 1 and classifies the remaining data as new spam and normal tweets in Stage 2. Therefore, the detection rate of known spam in Stage 1 determines the system’s performance quality. If spam is detected based on strict criteria in Stage 1, known spam will be detected well, but the FP rate will increase because uncertain normal tweets will also be classified as spam. Conversely, if spam is detected using loose criteria in Stage 1, much data will be transferred to Stage 2. In this case, unknown spam and normal tweets can be detected well in Stage 2; however, the rate of detecting known spam may decrease. Our framework is designed to be adaptable owing to intensity adjustability during Stage 1. A cost-based DT technique is used to adjust the DT at this stage. This method sets different costs for misclassification errors and then minimizes their sum [62,63]. Table 1 presents these asymmetric costs.

Cost-based classification uses an asymmetric cost matrix, as presented in Table 1. In the Stage 1 DT, known spam, which is small in number, is considered positive, and all other data are considered negative. There are two types of misclassifications—misclassification of known spam as negative and that of unknown spam as positive, with corresponding costs of $C\left(p,\;\;n\right)$ and $C\left(n,\;\;p\right)$, respectively. The goal of cost-based detection is to minimize the misclassification cost. Errors must be avoided as much as possible by setting $C\left(p,\;n\right)$ higher than $C\left(n,\;p\right)$ to maximize the detection accuracy in Stage 1. In contrast, if the detection accuracy needs to be maximized in Stage 2, $C\left(n,\;p\right)$ should be set higher than $C\left(p,\;n\right)$, sending more uncertain data to Stage 2. The cost $C\left(p,\;p\right)$ corresponding to true positives (TPs), i.e., when spam is detected as spam, and $C\left(n,\;n\right)$ corresponding to true negatives (TNs), when normal samples are detected as normal, must be set to 0. This is because misclassification increases the cost [62].

After all costs are set, the class of the given sample is determined by calculating the probability that it belongs to each class. The probability that input x belongs to class j can be determined as follows [62]:(6)$$P\left(j|x\right)=\frac{1}{\sum_{i}1}\sum_{i}P(j|x,M_{i}),$$
where i ranges between 1 and m, and m denotes the number of new resamples. The number of inputs for each resampling is n. Suppose $S_{i}$ denotes a resampling with n inputs. Then, $M_{i}$ is a model generated by applying a DT to $S_{i}$. The risk arising when x belongs to class k is defined as follows [64]:(7)$$R\left(k|x\right)=\sum_{j}P\left(j|x\right)C\left(k,j\right).$$

The aforementioned risk must be minimized when data are assigned to a class. Thus, x is assigned to the class k if k satisfies the following equation:(8)$${argmin}_{k}R(k|x).$$

Because there are two classes for the DT during Stage 1, k is one of two classes—known spam tweets and tweets that are not known to be spam.

In cost-based classification, the misclassification cost is inversely related to the error. Therefore, in this study, when the current tweet spam stream is mostly known spam, $C\left(p,\;n\right)$ is set to be higher than $C\left(n,\;p\right)$ to detect as much known spam as possible in Stage 1. Conversely, when the number of unknown spam is higher than that of known spam, $C\left(n,\;p\right)$ is set to be higher than $C\left(p,\;n\right)$ to detect obvious known spam in Stage 1 and check for ambiguous tweets in the anomaly detection phase during Stage 2. Thus, the proposed method can be adapted to the conditions of the input tweets by adjusting the cost of the pertinent error to be greater than those of other errors. Further, this effect increases with the cost difference, which allows the system to respond to extreme situations. Figure 4 concisely illustrates spam detection using a DT during Stage 1 by adjusting the cost.

## 4. Results

This study proposes a framework that outperforms existing methods in terms of detecting known and unknown spam based on a combination of anomaly detection and DT. A real-world dataset is used to validate the framework in an environment similar to a real application scenario, with the data being preprocessed first using a clustering technique to extract new spam tweets. Section 4.1 provides a detailed description of the dataset used in this study and its metrics. Section 4.2 evaluates the proposed framework’s effectiveness in detecting new spam. Section 4.3 examines the cost-based scalable framework’s performance.

### 4.1. Dataset Description and Evaluation Metrics

We used a dataset released by Chen et al. [25] after analyzing 600 million tweets. This dataset has been used in many studies because it is publicly available and consists of data collected from a real-world environment. Among the datasets released, we used the one with a spam-to-normal tweet ratio of 5:95 for validation in real-world conditions. The dataset contains 100,000 tweets, 12 statistical features, and account information associated with the tweets. Table 2 presents these features in detail.

This study’s primary purpose is to detect newly emerging types of spam tweets. Since spam tweets are grouped into only one class in our dataset, it is challenging to distinguish between known and unknown spam tweets. Therefore, they are first split into four clusters using a self-organizing map [65]. Clusters 1 and 2 represent the majority clusters, containing 1829 and 1562 spam tweets, respectively, of the 5000 spam tweets in aggregate. The tweets are assumed to be known spam tweets. Clusters 3 and 4 contain 861 and 748 spam tweets, respectively, and exhibit unique characteristics compared to Clusters 1 and 2.

Based on the spam tweets classified in this manner, Datasets 1 and 2 are constructed. The 95,000 normal tweets constituting Datasets 1 and 2 are identical. However, in Dataset 1, Clusters 1, 2, and 4 are categorized as known spam and Cluster 3 as unknown spam. The tweets in Cluster 3 in Dataset 1 exhibit distinctly different characteristics compared to other spam—despite their corresponding accounts being newly created, they are observed to exhibit active tweeting activity. In order words, Cluster 3 can be regarded to represent a previously unknown type of spammer that seeks to spread spam while acting like a normal user. In contrast, in Dataset 2, Clusters 1, 2, and 3 are classified as known spam and Cluster 4 as unknown spam. Cluster 4 also exhibits different characteristics compared to other spam. Therefore, we can consider Cluster 4 as unknown spam in Dataset 2. Table 3 provides detailed descriptions of the datasets.

We use precision, recall, and F-measure as the evaluation metrics. These metrics are calculated based on the basic TP, FP, and false negative (FN) values. Recall is a numerical value that represents the degree to which a class is correctly classified. Precision refers to the precision of detection and is defined to be the probability that the data point belongs to the class in which it was classified. The F-measure is the harmonic mean of precision and recall and represents overall performance. Precision, recall, and F-measure are calculated as follows:(9)$$Precision=\frac{TP}{TP+FP},Recall=\frac{TP}{TP+FN},F-Measure=\frac{2\cdot Recall\cdot Precision}{Recall+Precision}.$$

The primary goal of this study is to enhance the spam detection rate while simultaneously reducing the FP rate, i.e., the rate at which normal tweets are incorrectly classified as anomalies. Here, the detection rate refers to the proportion of total spam messages that are accurately classified by our framework. This can be expressed by the formula TP/(TP + FN). On the other hand, the FP rate refers to the proportion of normal tweets that are incorrectly classified as positive. This can be also represented by the formula FP/(FP + TN).

### 4.2. Performance Comparison of the Proposed Framework

This section compares the results of Stages 1 and 2 of the proposed method with those of conventional methods to examine the effectiveness of the combination of DT and anomaly detection. The effect of cost adjustment is examined in the next section.

Datasets 1 and 2, as described in the previous section, are used to compare the proposed framework’s performance with those of other methods. Our framework initially utilizes a DT and subsequently employs anomaly detection techniques, such as autoencoders and one-class SVMs. These models are implemented using scikit-learn and keras. CART-based DT is configured with a maximum depth of 10. For the autoencoder, excluding the input layer, four hidden layers are established during the encoding process, with 16, 8, 4, and 2 neurons, respectively. For the decoding process, excluding the output layer, three hidden layers are defined, with 4, 8, and 16 neurons, respectively. The activation function ‘elu’ is employed, along with ‘Adam’ as the optimizer and ‘MSE’ as the loss function. The number of epochs is set to 100. For the one-class SVM, the ‘RBF’ kernel is utilized, and the regularization parameter is set to 0.1.

The proposed framework distinguishes between known and unknown spam. Hence, its performance is compared with those of misuse detection-based ML methods. For the comparison, naïve Bayes, a statistical method; RF, an ensemble of several DTs; partial decision trees (PART), a DT-based system that obtains rules from partial decision trees and bagging; and LightGBM, a set of DTs connected through boosting, are considered. Further to this, the performances of the one-class SVM and autoencoder when used independently in isolation are also compared. The algorithms are implemented using Weka [66], scikit-learn, and Keras; and training and classification are performed using 5-fold cross-validation.

Table 4 presents the precision, recall, and F-measure scores of our framework while classifying normal tweets, known spam tweets, and unknown spam tweets based on Datasets 1 and 2. The framework always applies the same DT in Stage 1, but uses either the one-class SVM or autoencoder for anomaly detection in Stage 2. The results obtained using the DT and one-class SVM are presented in the upper part of Table 4, labeled “DT–SVM”, and those obtained using the DT and autoencoder are presented in the lower part of Table 4, labeled “DT–Autoencoder”. Despite slight differences in the detection results for the three types of tweet data, the results on Dataset 1 are observed to be better than those of Dataset 2. This difference in performance can be attributed to the higher degree if differentiation between the characteristics of unknown spam tweets in Dataset 1. On both Datasets 1 and 2, the results for the normal tweets were better than those for the two types of spam. This difference can be attributed to the greater ease of learning normal data, which accounted for most of the data. The precision and F-measure are greater than or equal to 0.9 for known spam, exhibiting good results on both datasets. Recall, which directly represents the spam detection rate, is observed to be in the high 0.8 range, indicating good detection performance. The unknown spam detection performance using anomaly detection is not better than that of known spam because of the nature of anomaly detection. Furthermore, using the autoencoder leads to better performance than using the one-class SVM. When both DT and autoencoder are used, 87.8% of the unknown spam is detected in Dataset 1 and 85.4% in Dataset 2, indicating the viability of responding to emerging types of spam using the proposed system.

Figure 5 and Figure 6 compare the performance of our framework with those of conventional methods in terms of detection rate, with comparisons corresponding to known and unknown spam, respectively. The charts are created using Datawrapper, a visual analytics tool [67]. Figure 5 and Figure 6 reveal that, despite slight variations across algorithms, the results are better on Dataset 1. In Figure 5, LightGBM exhibits the highest detection rate for known spam, with a detection rate of 91.8% on Dataset 1, followed by our method and RF. A similar trend is observed on Dataset 2. The detection rate of known spam achieved by our method is excellent and is not significantly different from that achieved by LightGBM, which is known to perform well among misuse detection methods. Our method outperformed traditional misuse methods, such as naïve Bayes, PART, and RF. The one-class SVM and autoencoder are trained with only normal data because they are anomaly detection methods. Consequently, they exhibit a lower detection rate than our method for known spam, confirming that anomaly detection cannot be relied upon on its own, despite being effective at detecting new types of spam. Figure 6 compares the detection rates for unknown spam and reveals that the DT–SVM and DT–Autoencoder methods detect this type of spam better than the traditional ones. Because the traditional misuse detection methods do not learn unknown types of spam, their detection rate is low. Our method detects unknown spam better than methods that use anomaly detection in isolation. The proposed method performs better because, as previously explained, subsets with similar types of data obtained using DT are learned individually and then used for detection.

A comparison of spam detection rates reveals that the proposed method responds well to known and unknown spam. However, the FP rate is as important as the detection rate in tasks such as spam detection. The FP rate at which normal tweets are categorized as spam does not pose a security threat because spam tweets are not considered non-spam tweets. However, a high FP rate incurs high overhead, which burdens the system, and frequent FPs degrade confidence of security system users. Therefore, the FP rates of our framework and traditional methods are compared for known and unknown spam in Figure 7 and Figure 8. The misuse detection method exhibits a relatively lower FP rate because both normal and attack data are learned. In contrast, anomaly detection yields a high FP rate because only normal data are learned. DT–SVM and DT–Autoencoder are hybrid methods that combine DT and anomaly detection, and they are instrumental since they exhibit the lowest FP rates for known and unknown spam tweets, while responding well to unknown spam.

### 4.3. Performance Comparison Based on Cost Changes

The proposed framework can respond to given situations by adjusting the costs during the DT stage. When the occurrence of existing spam types is high, $C\left(p,\;\;n\right)$ is set to be high. In contrast, if there are many new types of spam, $C\left(n,\;\;p\right)$ is set to be high. Figure 9 and Figure 10 depict the change in detection rate for known and unknown spam on Datasets 1 and 2. The left sides of Figure 9 and Figure 10 represent cases when known spam types are more common, and *C*(*p*, *n*) is set to 5, 10, and 15. Here, *C*(*n*, *p*) is set to 1. As *C*(*p*, *n*) is set to be higher, the detection rate for known spam increases as it is harder to miss. The highest detection rate was observed when Dataset 1 was utilized with DT–Autoencoder. Setting *C*(*p*, *n*) to 15 resulted in a detection rate of 92.8% for known spam, which is 2.1% higher than the detection rate of 90.7% achieved by systems that do not adjust the costs. This indicates that increasing *C*(*p*, *n*) can effectively enhance the response to spam in scenarios where known spam is prevalent. In contrast, if *C*(*n*, *p*) is set to 5, 10, and 15 and *C*(*p*, *n*) to 1, the detection rate for unknown spam improves slightly, but not as much as that when *C*(*p*, *n*) is increased. When employing DT–Autoencoder on Dataset 1 with *C*(*n*, *p*) set to 15, the detection rate for unknown spam reaches 88.5%. This is slightly higher than the 87.8% detection rate for unknown spam in systems that do not adjust the costs. This improvement can be attributed to the cost of the DT process—the detection of unknown spam relies on anomaly detection. Our proposed scalable framework is more effective when the threat of existing spam is greater than that of new spam and when the detection of known spam is improved.

## 5. Conclusions

This study proposes a framework designed to detect known and unknown spam tweets effectively. The proposed method combines DT with either a one-class SVM or an autoencoder and achieves higher detection rates than most conventional methods for both known and unknown spam. For unknown spam, its detection rate is observed to be better than that of traditional anomaly detection methods. Furthermore, its detection rate for known spam is not significantly lower than that of LightGBM, which is the best-performing conventional misuse detection method. Our method responds well to unknown spam without a significant performance change in detecting known spam when compared to the best misuse detection methods. Moreover, it is adaptable to environmental changes by adjusting the costs to different errors in the DT stage. Experimental results demonstrate that our method responds better than existing alternatives, particularly in situations with a high volume of known spam attacks.

In this study, we utilize features that can be easily extracted from Twitter, enhancing the applicability of the proposed method. However, if the challenge of collection of higher-dimensional features can be addressed effectively, their use can further enhance accuracy. To this end, feature collection based on recently developed language models should be investigated to enable more in-depth text analysis. Additionally, the correlations between accounts should also be analyzed further to increase accuracy. Further to this, although the dataset used in this study is widely used for research on spam tweet detection, it does not reflect current trends because it was released in 2015. Further research should be conducted on reliable open datasets that reflect current trends more accurately.

The significance of the techniques proposed in this study lies in their ability to respond to emerging threats effectively. There is a demand for methods capable of responding to new threats in various security fields beyond social networks, including spam emails, malware, and malicious websites. Expanding the application scope of this research to other domains could contribute significantly to various security fields.

## Figures and Tables

**Figure 1 sensors-24-02263-f001:**
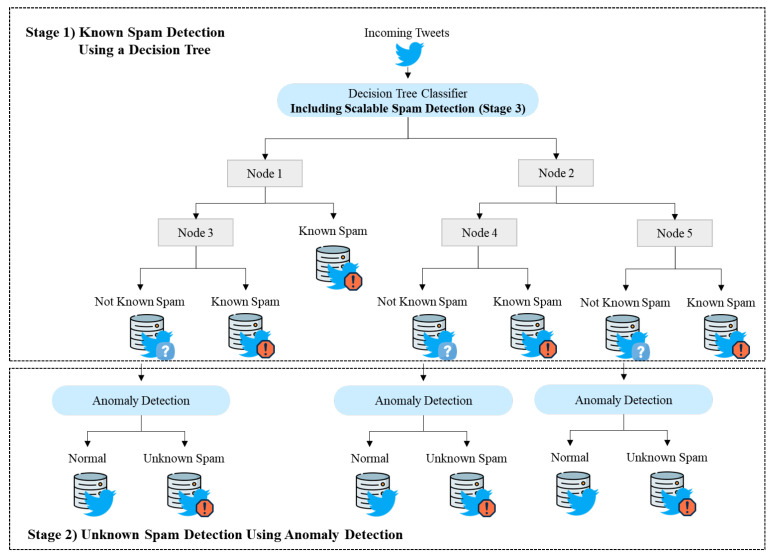
Overall decision-making process of the proposed framework.

**Figure 2 sensors-24-02263-f002:**
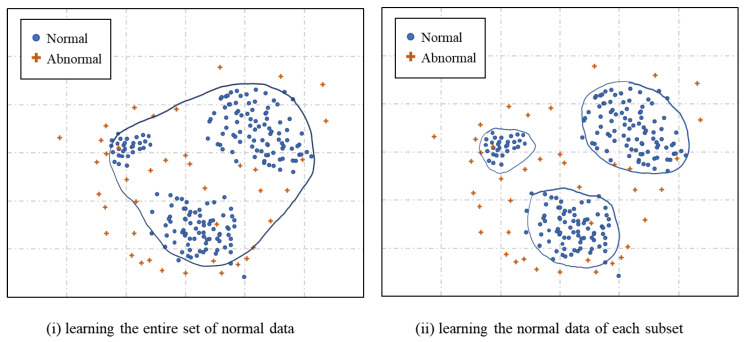
Comparison of anomaly detection based on the entire normal dataset and anomaly detection based on individual subsets of normal data.

**Figure 3 sensors-24-02263-f003:**
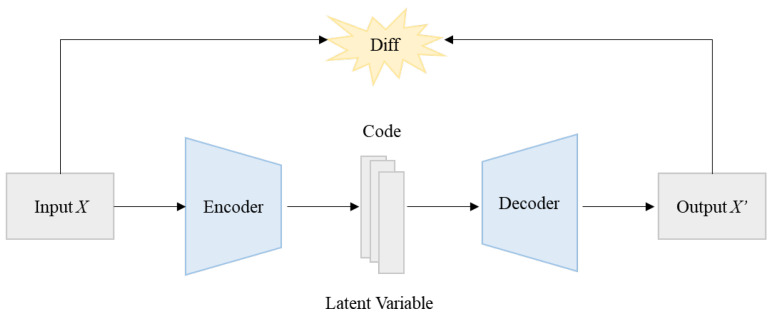
Autoencoder-based unsupervised anomaly detection.

**Figure 4 sensors-24-02263-f004:**
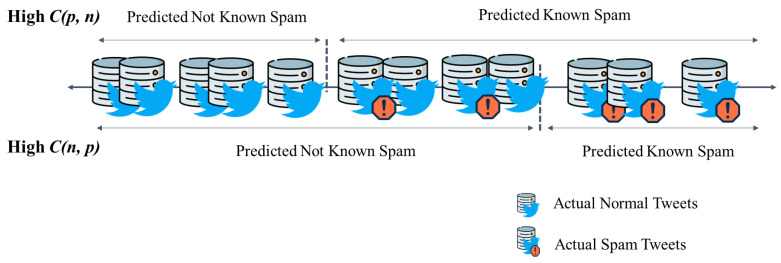
Example of spam detection by adjusting the cost while using a DT.

**Figure 5 sensors-24-02263-f005:**
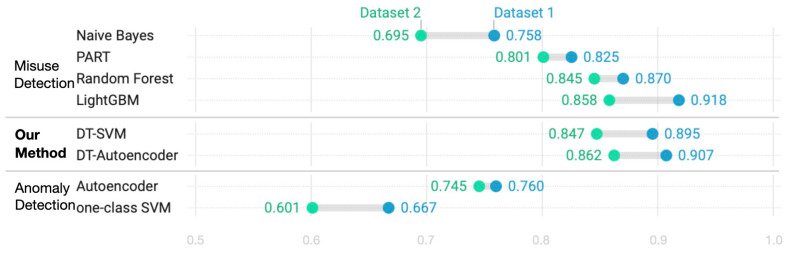
Comparison of detection rates for known spam tweets. Results for (**top**) misuse detection methods, (**middle**) our method, and (**bottom**) anomaly detection methods.

**Figure 6 sensors-24-02263-f006:**
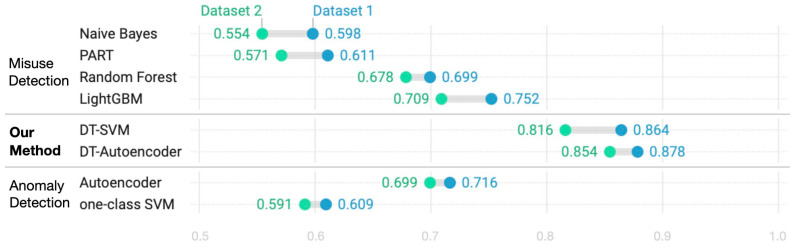
Comparison of detection rates for unknown spam tweets. Results corresponding to (**top**) misuse methods, (**middle**) our method, and (**bottom**) anomaly detection methods.

**Figure 7 sensors-24-02263-f007:**
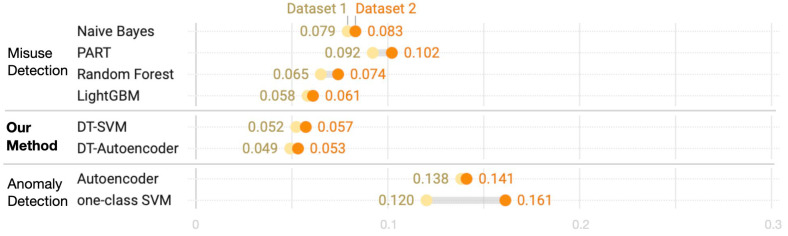
Comparison of FP rates for known spam tweets.

**Figure 8 sensors-24-02263-f008:**
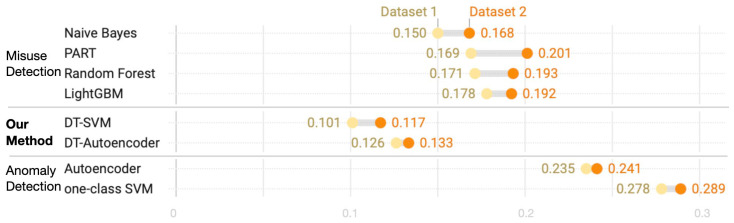
Comparison of FP rates for unknown spam tweets.

**Figure 9 sensors-24-02263-f009:**
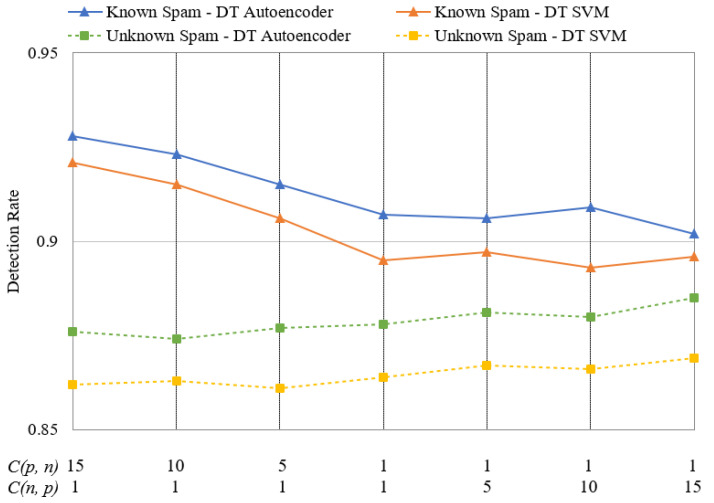
Changes in spam detection rate with respect to changes in *C*(*p*, *n*) and *C*(*n*, *p*) on Dataset 1.

**Figure 10 sensors-24-02263-f010:**
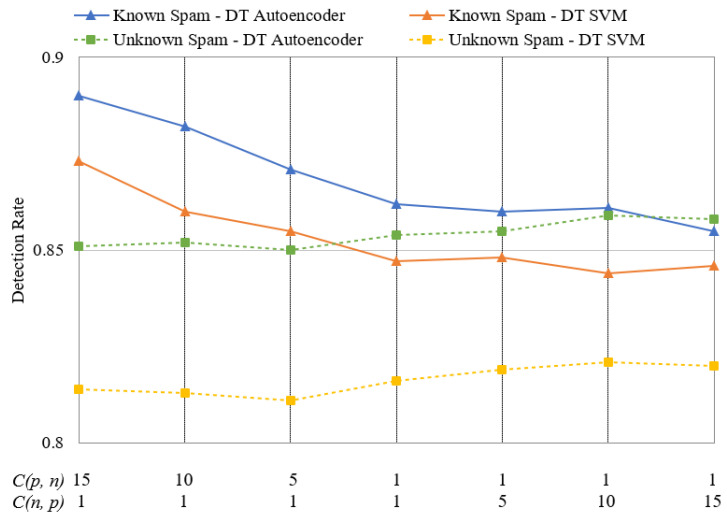
Changes in the spam detection rate with respect to changes in *C*(*p*, *n*) and *C*(*n*, *p*) on Dataset 2.

**Table 1 sensors-24-02263-t001:** Asymmetric misclassification cost matrix.

	Actual Positive	Actual Negative
Predicted Positive	C(p, p)	$C\left(n,\;p\right)$
Predicted Negative	$C\left(p,\;n\right)$	$C\left(n,\;n\right)$

**Table 2 sensors-24-02263-t002:** Feature descriptions.

Type	Feature Name	Feature Description
Tweet-Based Features	no_userfavourites	Number of likes received by the user who posted the tweet
	no_lists	Number of lists added by the user who posted the tweet
	no_tweet	Number of tweets posted by the user who posted the tweet
	no_retweet	Number of retweets of the tweet
	no_hashtags (#)	Number of hashtags included in the tweet
	no_usermention (@)	Number of user mentions included in the tweet
	no_url	Number of URLs included in the tweet
	no_char	Number of characters in the tweet
	no_digit	Number of numerical digits in the tweet
Account-Based Features	account_age	Number of days between the date the account was first created and the date the most recent tweet was posted
	no_follwer	Number of followers of the user who posted the tweet
	no_following	Number of accounts the user who posted the tweet follows

**Table 3 sensors-24-02263-t003:** Composition of Datasets 1 and 2.

Cluster	Main Characteristics	Number	Dataset 1	Dataset 2
Cluster 1	Recent account creation date Extremely small number of account followers; Small number of accounts followed Small number of tweets posted and likes received Almost no retweets Short tweet length, and tweets do not contain many numbers	1829	Known Spam	Known Spam
Cluster 2	Old account creation date Small number of followers; Extremely small number of followed accounts Small numbers of likes received and tweets posted Almost no retweets Short tweet length, and tweets do not contain many numbers	1562	Known Spam	Known Spam
Cluster 3	Recent account creation date Many follower and followed accounts Many likes received and tweets posted Many retweets Long tweet length, and tweets contain numbers	861	Unknown Spam	Known Spam
Cluster 4	Old account creation date Many follower and followed accounts Many likes received and tweets posted Almost no retweets Average tweet length, and tweets contain numbers	748	Known Spam	Unknown Spam

**Table 4 sensors-24-02263-t004:** Overall performance of the proposed framework.

Framework	Tweet Type	Dataset 1			Dataset 2		
		Precision	Recall	F-Measure	Precision	Recall	F-Measure
DT–SVM	Known Spam	0.945	0.895	0.920	0.937	0.847	0.890
	Unknown Spam	0.873	0.864	0.868	0.860	0.816	0.837
	Normal	0.994	0.997	0.995	0.992	0.997	0.994
DT–Autoencoder	Known Spam	0.949	0.907	0.928	0.942	0.862	0.900
	Unknown Spam	0.897	0.878	0.887	0.879	0.854	0.867
	Normal	0.995	0.997	0.996	0.993	0.997	0.995

## Data Availability

The data are available upon reasonable request from the corresponding author.

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
