# Peer review of "Scalable Learning Framework for Detecting New Types of Twitter Spam with Misuse and Anomaly Detection"

_sensors, 2024, doi:10.3390/s24072263_

Round 1

Reviewer 1 Report

Comments and Suggestions for Authors

This study proposes a framework that utilizes a decision tree, SVM, and autoencoder to effectively detect new spam to avoid the problem of low detection rate and high FP rate.

 However, Some essential viewpoints need to be clarified to reveal the value of this research.

1.  everything can’t be spam on Twitter or X. How to define unknown spam or new spam, authors must first have a clear definition.  Currently, the platform only considers messages and posts spam when they worsen the other user experience. So in this research what is the unknown spam?

2.  Real-world unknown spam cannot be simulated using the existing spam dataset. In particular, there is the "Twitter Spam Drift" phenomenon. It will constantly change, rather than being a static category like the experiment here. We hope that the author can have a more solid solution and simulation in this part.

3.  SL and USL should not be used as a type of algorithm. It is a general term for describing machine learning training. It should be replaced by a more precise algorithm name.

4.  The author specifically mentioned the improvement of the false positive rate but did not explain its importance. Why not evaluate by the false negative rate?

In addition, the article also contains many writing errors. Authors should Authors should check and read proofing repeatedly.

For instance:

Line465: We used a dataset released by Chen et al. [23] after analyzing 600 million tweets. The reference should be [22]

 Finally, Twitter has now changed its name to “X”. The author may update the latest information

Author Response

1. Summary

Thank you very much for taking the time to review this manuscript. Revisions reflecting Reviewer 1’s comments have been highlighted in yellow in the manuscript, and those reflecting Reviewer 2’s comments have been highlighted in light blue. Moreover, revisions reflecting the comments of both reviewers have been highlighted in green. Grammatical revisions have been highlighted in gray. Please find our detailed responses below

2. Point-by-point response to Comments and Suggestions for Authors

Comments 1: everything can’t be spam on Twitter or X. How to define unknown spam or new spam, authors must first have a clear definition.  Currently, the platform only considers messages and posts spam when they worsen the other user experience. So in this research what is the unknown spam?

Response 1: Thank you for your comment. We acknowledge the lack of a clear definition of spam and the term 'unknown spam’ in the original manuscript. In this study, 'Unknown spam' refers to content that does not fit within previously defined categories of spam and uses new techniques or methodologies to deceive ordinary users or systems. Such spam is not easily identifiable by traditional spam filtering techniques, and we propose a new approach to address the continuously evolving strategies of emerging spam.

We have added an explanation to the Introduction to address this. As pointed out, the definition of unknown or new spam is crucial; hence, we have detailed this in a new paragraph, which can be found in lines 48–63 on page 2.

Further, to convey the definition of known and unknown spam more clearly, we have included additional explanations in Materials and Methods. This information can be found between line 253-256 on page 5.

Comments 2: Real-world unknown spam cannot be simulated using the existing spam dataset. In particular, there is the "Twitter Spam Drift" phenomenon. It will constantly change, rather than being a static category like the experiment here. We hope that the author can have a more solid solution and simulation in this part.

Response 2: We greatly appreciate the reviewer's insight into the dynamic nature of spam, particularly highlighting the "Twitter Spam Drift" phenomenon. We have utilized the Twitter spam public dataset widely used in various studies in our experiments. As noted, this dataset contains only spam that has already been collected. To define what should be considered unknown spam within this dataset clearly, we employed clustering techniques to isolate a small group of spam with unique characteristics. These were then defined as a new form of unknown spam.

We had acknowledged this process within Results in the original manuscript, but realize the explanation may have been insufficient for clear understanding. To address this, we have added a more detailed description of the process to Section 4. This additional information can be found in lines 515–521 and 525–538 on page 12, and lines 529-532 on page 13..

Comments 3: SL and USL should not be used as a type of algorithm. It is a general term for describing machine learning training. It should be replaced by a more precise algorithm name.

Response 3: We thank the reviewer for their critical observation regarding the use of the terms Supervised Learning (SL) and Unsupervised Learning (USL) in our manuscript. As noted in the comment, since SL and USL denote types of learning methods, we have replaced them with terms related to detection. SL has been changed to misuse detection, which detects known attacks, and USL has been updated to anomaly detection, which identifies data deviating from a previously established norm.

Following these revisions, several changes have been made throughout the manuscript. These amendments can be found in the following locations: Page 2, lines 82, 83; 85, 86; 90-94; 95, 96. Page 3, lines 105-109; 119, 120; 126, 128; 140, 141; 148. Page 4, lines 151; 154-155. Page 5, lines 206–207; 210-211; 220-224; 228-230; 241-244. Page 7, line 324. Page 9, line 389. Page 14, line 565. Page 16, line 631.

Comments 4: The author specifically mentioned the improvement of the false positive rate but did not explain its importance. Why not evaluate by the false negative rate?

Response 4: We appreciate the reviewer's comment on the emphasis placed on the improvement of the false positive rate in our study without adequately explaining its significance. Reducing false positives is very important in spam detection applications because a high false positive rate could lead to the unwarranted blocking of legitimate messages, which in turn could significantly degrade user experience.

We acknowledge the lack of explanation regarding the importance of improving the false positive rate and have added supplementary information to address this. The importance of minimizing the false positive rate has been discussed in the Introduction, which can be found lines 105–108 on page 3. Additionally, while describing experimental metrics, we have included a precise definition of the false positive rate. This information can be found in lines 543–547 on page 13.

We appreciate the reviewer's question regarding the absence of an explicit evaluation based on the false negative rate in our study. Our decision to not separately report the false negative rate was guided by our analytical framework, which focuses on the detection rate as the primary metric. Since the detection rate and false negative rate are inversely related within a binary classification system, understanding one metric offers implicit insights into the other. Specifically, in our context, the sum of the detection rate and the false negative rate is always equal to 1, meaning that an improvement in the detection rate directly indicates a reduction in the false negative rate. Therefore, we have added a discussion of the detection rate to the subsection dealing with experimental metrics. This clarification aims to provide a thorough understanding of our decision to focus on the detection rate as the primary metric and its implicit representation of the false negative rate within our study's evaluation framework. This explanation can be also found in lines 543–547 on page 13.

Additional comments: In addition, the article also contains many writing errors. Authors should Authors should check and read proofing repeatedly.

For instance:

Line465: We used a dataset released by Chen et al. [23] after analyzing 600 million tweets. The reference should be [22]

Response 4: We thank the reviewer for helping us improve the quality of our manuscript. Based on the author’s comment, we have corrected the incorrect citation. This can now be verified in line 506 on page 12. Additionally, we corrected the ordering of references in two other instances, which can be checked in line 98 on page 2, line 453 on page 10, and line 476 on page 11.

To enhance the overall quality of the English in the manuscript, we have conducted a comprehensive review and correction. Initially, we had the manuscript proofread by native speakers from Editage before our first submission. However, acknowledging the feedback regarding the unsatisfactory quality of English, we sought a complete re-proofreading from Editage prior to submitting our revision. We have highlighted the revised sections of the manuscript in gray. Documentation of this proofreading service will also be submitted for verification.

Additional comments: Finally, Twitter has now changed its name to “X”. The author may update the latest information

Response 4: We appreciate the reviewer's suggestion to update the terminology in our manuscript to reflect Twitter's name change to "X." It is important to note that our research specifically utilizes data from the platform when it was known and widely referred to as Twitter. This dataset encompasses spam content identified during that period, making the term "Twitter" not only historically accurate but also relevant to the context of our study.

To maintain clarity and ensure the accuracy of our analysis, we have chosen to continue referring to the platform as "Twitter" throughout our manuscript. This decision is rooted in our intention to provide readers with a clear understanding of the data's origin and the specific context within which our research was conducted. We have added an explanation to the Introduction regarding our decision to use the name "Twitter" in our manuscript. This clarification can be found in lines 60–63 on page 2.

Reviewer 2 Report

Comments and Suggestions for Authors

Section – wise report:

1.      Introduction:

The introduction could benefit from a more detailed historical context of Twitter spam evolution, providing a richer background for the significance of the new scalable learning framework. The introduction broadly mentions spam detection challenges but lacks specificity in discussing previous limitations and how the proposed framework addresses these directly.

2.      Literature Review:

There is a lack of critical analysis comparing the strengths and weaknesses of existing methods, which would provide a stronger rationale for the proposed framework. The review could be updated to include more recent developments in the field, ensuring the research's relevance and context are grounded in the latest findings.

3.      Methodology:

The methodology section could benefit from a more detailed description of the data collection process, including how the Twitter data was sourced and the criteria for labeling spam. While the use of anomaly detection and decision trees is innovative, the paper lacks detailed information on the parameters and settings of these algorithms, which is crucial for reproducibility. The methodology section could improve by including a discussion on validation techniques and how the model's performance was evaluated against known benchmarks.

4.      Results:

The results are presented with quantitative data but lack a deeper interpretation of what these findings mean in the context of Twitter spam detection.

The section could be enhanced by incorporating more visual representations of the results, such as charts or graphs, to aid in understanding the framework's performance.

5.      Conclusion:

Any research has limitations, and briefly acknowledging these in the conclusion can provide a balanced view of your work. The practical implications of the research are mentioned in broad terms; detailing how the framework can be implemented or adapted by social media platforms would add value. The conclusion lacks a strong call to action for researchers or practitioners in the field, missing an opportunity to inspire further investigation or adoption of the proposed framework.

Author Response

1. Summary

Thank you very much for taking the time to review this manuscript. Revisions reflecting Reviewer 1’s comments have been highlighted in yellow in the manuscript, and those reflecting Reviewer 2’s comments have been highlighted in light blue. Moreover, revisions reflecting the comments of both reviewers have been highlighted in green. Grammatical revisions have been highlighted in gray. Please find our detailed responses below.

2. Point-by-point response to Comments and Suggestions for Authors

Comments 1: The introduction could benefit from a more detailed historical context of Twitter spam evolution, providing a richer background for the significance of the new scalable learning framework. The introduction broadly mentions spam detection challenges but lacks specificity in discussing previous limitations and how the proposed framework addresses these directly.

Response 1: We appreciate the reviewer’s suggestion to enrich the introduction. Firstly, we have added a discussion on the limitations of existing techniques. Specifically, we have elaborated on the vulnerability of spam detection methods based solely on misuse detection to new attacks, and the low detection accuracy of those employing anomaly detection. These points can be found in lines 85–86 and lines 93–94 on page 2.

Further, we have revised the explanation of how the proposed technique addresses these issues. Our approach first filters known attacks using misuse detection, and then performs anomaly detection as a secondary measure. This dual-step process underscores its comprehensive strategy to combat spam, enhancing both the detection of known threats and the identification of novel, anomalous behaviors. This detailed explanation has been added in lines 118–120 on page 3.

Comments 2: There is a lack of critical analysis comparing the strengths and weaknesses of existing methods, which would provide a stronger rationale for the proposed framework. The review could be updated to include more recent developments in the field, ensuring the research's relevance and context are grounded in the latest findings.

Response 2: Acknowledging the reviewer's insightful feedback, we have taken steps to enhance the manuscript by incorporating a critical analysis of the strengths and weaknesses of existing spam detection methods.

This updated review includes an in-depth discussion on the limitations of current approaches, such as the poor adaptability of misuse detection systems to emerging spam tactics and the challenges of maintaining high accuracy in anomaly detection methods amidst evolving spam patterns. In relation to this, the updated content can be found on pages 4 and 5, in lines 203–206, and in lines 220–224, lines 231–234, and lines 241–243 on page 5.

Additionally, we have reviewed recent studies in this field based on cutting-edge deep learning techniques or large language models. This related content can be found in lines 177–179 and 200–201 on page 4, and lines 230–231 on page 5.

Comments 3: The methodology section could benefit from a more detailed description of the data collection process, including how the Twitter data was sourced and the criteria for labeling spam. While the use of anomaly detection and decision trees is innovative, the paper lacks detailed information on the parameters and settings of these algorithms, which is crucial for reproducibility. The methodology section could improve by including a discussion on validation techniques and how the model's performance was evaluated against known benchmarks.

Response 3: Acknowledging the reviewer’s constructive feedback, we have significantly enhanced the methodology section of our manuscript to provide a clearer, more comprehensive description of our experiments.

To address the concerns regarding the lack of detailed information on the algorithms used, we have included a thorough description of the parameters and settings for both the anomaly detection techniques and decision trees implemented in our study. This content has been updated and can be found in lines 554–563 on page 14. By providing these details, we aim to enhance the reproducibility of our research, enabling other scholars to replicate or build upon our work with precision.

Comments 4: The results are presented with quantitative data but lack a deeper interpretation of what these findings mean in the context of Twitter spam detection.

The section could be enhanced by incorporating more visual representations of the results, such as charts or graphs, to aid in understanding the framework's performance.

 Response 4: We appreciate the reviewer's observation regarding the presentation of our results and their suggestion to enrich the interpretation and visualizations of our findings.

To compare the experimental results of our framework with those of existing techniques more clearly, we have revised our figures extensively. Utilizing the Datawrapper tool, which is known for its ability to provide visually appealing charts, we have reconstructed the comparison charts for detection rate and false positive rate. The revised figures can be found in Figures 5-8. We hope that this enhances the clarity and impact of our results, making it easier for readers to understand the advancements achieved by our framework within the context of Twitter spam detection.

Comments 5: Any research has limitations, and briefly acknowledging these in the conclusion can provide a balanced view of your work. The practical implications of the research are mentioned in broad terms; detailing how the framework can be implemented or adapted by social media platforms would add value. The conclusion lacks a strong call to action for researchers or practitioners in the field, missing an opportunity to inspire further investigation or adoption of the proposed framework.

Response 5: In response to the reviewer's valuable feedback, we have revised the conclusion of our manuscript to provide a more balanced and comprehensive perspective of our work, acknowledging its limitations, detailing practical implications, and incorporating a strong call to action for both researchers and practitioners in the field.

Firstly, we have explicitly addressed the limitations of our research. Our research is highly versatile, as it is based on easily collectable features. However, its performance can potentially be further improved by utilizing more complex features and algorithms. In particular, employing deep learning techniques for text analysis, which is currently an active topic of research, could not only enhance performance but also become a new research topic in itself. We have added a description related to this in lines 674–679 on page 19.

Moreover, we have expanded the discussion on the practical implications of our research, offering detailed insights into how our framework can be implemented in other security fields. The content providing further research directions for scholars can be found in lines 683–687 on page 19.

We hope that these revisions ensure that our conclusion offers a clear, balanced, and inspiring overview of our research, its implications, and the potential for future advancements in the field of spam detection. Thank you for your comments.

Round 2

Reviewer 1 Report

Comments and Suggestions for Authors

1.     

In Response 4, the author mentioned:

[…Since the detection rate and false negative rate are inversely related within a binary classification system, understanding one metric offers implicit insights into the other…]

The detection rate and false negative rate are not inversely related.

Please refer to the below paper which will review the definitions of Recall, FPR, and FNR.

Twitter Spam Detection: A Systematic Review

https://arxiv.org/pdf/2011.14754

Also, in line 595:

[…Figures 5 and 6 compare the performance of our framework with those of conventional methods in terms of detection accuracy, or detection rate…]

Based on the abovementioned reference, accuracy is not the same as detection rate.

The authors should recheck any misinterpretations of performance metrics in the paper.

2.

In line 545:

[Here, the detection rate refers to the proportion of total spam messages that are accurately classified by our framework. On the other hand, the FP rate refers to the proportion of normal tweets that are incorrectly classified as positive. ]

Detection rate is a confusing and misleading term. If we based on the definition on “Twitter Spam Detection: A Systematic Review”, it should be TP/(FN+FP+TN+TP)

Moreover, Refer to: Guidance on Terminology;

https://journals.sagepub.com/doi/pdf/10.1177/096914139400100118

[ …Detection rate (DR) and sensitivity are synonyms (the proportion of affected individuals with a positive test result).

An advantage of "detection rate" is that it avoids confusion as "sensitivity" has a different meaning in analytical biochemistry (the minimum detectable amount in an assay). Detection rate can be used in a different sense in cancer screening - as the number of screen positive individuals divided by the number screened. This is better described as the prevalence of screen positive cancers in the population.

False positive rate (FPR) (the proportion of unaffected individuals with positive results) is the complement of specificity (the proportion of unaffected individuals with negative results) or (100- FPR) expressed as a percentage. The advantage of using the term false positive rate is that Journal of Medical Screening 1994;1:76 (a) it is more easily understood and remembered, (b) it focuses attention on the group who will be offered further medical intervention, (c) a 10% false positive rate, for example, is twice as bad as one of 5%, whereas the corresponding specificity values of 90% and 95% conceal the difference.… ]

The detection rate has the meaning of the prevalence screen of spam on Twitter, which is less meaningful to evaluate the detection algorithm.

Therefore, I suggest using it as little as possible in the detailed content, unless you are talking about the overall picture. This is why there is no such metric in your experimental result.

For example, In line 543:

[The primary goal of this study is to enhance the spam detection rate while simultaneously reducing the FP rate]

It should be better replaced with “enhance the spam TP rate while simultaneously reducing the FP rate”, as the TPR is in contrast to FPR.

3.

Also, in the abovementioned reference, their review show:

[…Twitter spam detection is related to recall parameter with 23%, and F-measure comes next with 18%, precision with 17%, and accuracy 14%. …]

Hence, the section “Performance Comparison Based on Cost Changes” is a new viewpoint.

The authors can enhance this part of the discussion to reveal their innovation.  

4.

The title “Scalable Learning Framework for Detecting New Types of Twitter Spam with Anomaly Detection and Decision Trees”

This title is a little strange. The whole article does not mention the scalability part. Moreover, Decision Trees is a method of Anomaly Detection and should not be written together.

Author Response

1. Summary

Thank you very much for taking the time to review this manuscript. Revisions reflecting Reviewer 1’s comments have been highlighted in yellow in the manuscript. Please find our detailed responses below

2. Point-by-point response to Comments and Suggestions for Authors

Comments 1: In Response 4, the author mentioned:

[…Since the detection rate and false negative rate are inversely related within a binary classification system, understanding one metric offers implicit insights into the other…]

The detection rate and false negative rate are not inversely related.

Please refer to the below paper which will review the definitions of Recall, FPR, and FNR.

Twitter Spam Detection: A Systematic Review

https://arxiv.org/pdf/2011.14754

Also, in line 595:

[…Figures 5 and 6 compare the performance of our framework with those of conventional methods in terms of detection accuracy, or detection rate…]

Based on the abovementioned reference, accuracy is not the same as detection rate.

The authors should recheck any misinterpretations of performance metrics in the paper.

Comments 2: In line 545:

[…Here, the detection rate refers to the proportion of total spam messages that are accurately classified by our framework. On the other hand, the FP rate refers to the proportion of normal tweets that are incorrectly classified as positive. …]

Detection rate is a confusing and misleading term. If we based on the definition on “Twitter Spam Detection: A Systematic Review”, it should be TP/(FN+FP+TN+TP)

Moreover, Refer to: Guidance on Terminology;

https://journals.sagepub.com/doi/pdf/10.1177/096914139400100118

[ …Detection rate (DR) and sensitivity are synonyms (the proportion of affected individuals with a positive test result).

An advantage of "detection rate" is that it avoids confusion as "sensitivity" has a different meaning in analytical biochemistry (the minimum detectable amount in an assay). Detection rate can be used in a different sense in cancer screening - as the number of screen positive individuals divided by the number screened. This is better described as the prevalence of screen positive cancers in the population.

False positive rate (FPR) (the proportion of unaffected individuals with positive results) is the complement of specificity (the proportion of unaffected individuals with negative results) or (100- FPR) expressed as a percentage. The advantage of using the term false positive rate is that Journal of Medical Screening 1994;1:76 (a) it is more easily understood and remembered, (b) it focuses attention on the group who will be offered further medical intervention, (c) a 10% false positive rate, for example, is twice as bad as one of 5%, whereas the corresponding specificity values of 90% and 95% conceal the difference.… ]

The detection rate has the meaning of the prevalence screen of spam on Twitter, which is less meaningful to evaluate the detection algorithm.

Therefore, I suggest using it as little as possible in the detailed content, unless you are talking about the overall picture. This is why there is no such metric in your experimental result.

For example, In line 543:

[…The primary goal of this study is to enhance the spam detection rate while simultaneously reducing the FP rate…]

It should be better replaced with “enhance the spam TP rate while simultaneously reducing the FP rate”, as the TPR is in contrast to FPR.

Response 1 & 2:

First of all, I sincerely thank you for your response. I will address comments 1 and 2 together since they both relate to metrics.

It appears there was a lack of detailed explanation regarding the detection rate. In accordance with the review comments, we have made efforts to improve this.

Initially, the detection rate used in this paper represents the proportion of actual spam (positive) that was predicted as spam, which is calculated as TP/(FN + TP). When spam is considered Positive, the recall for spam was what we referred to as the detection rate. Therefore, the reason I mentioned that it was related to FN in our last response was because the sum of FN/(TP+FN) and TP/(TP+FN) equals 1. We apologize for any confusion caused by the insufficient explanation.

Predicted Negative(Normal)

Predicted Postive(Spam)

Actual Negative(Normal)

TN

FP

Actual Positive(Spam)

FN

TP

The False Positive Rate mentioned in the paper is also defined as the proportion of all normal data that was incorrectly classified as spam, which is calculated as FP/(FP + TN).

We recognize that many papers in the field of spam detection, such as the one linked below, employ metrics like detection rate and false positive rate without extensive elucidation. This practice stems from an implicit understanding among researchers familiar with these concepts. However, considering the potential for interdisciplinary application of our findings and the importance of methodological transparency, we have chosen to include a more detailed explanation of these metrics. Our aim is to ensure that our research is accessible not only to experts in spam detection but also to a broader audience that may benefit from our work.

https://www.sciencedirect.com/science/article/pii/S0140366413001047

We fully appreciate the feedback received and understand the importance of clarity, especially for scholars new to the field of spam detection. Consequently, we have added a detailed explanation of the detection rate in the introduction of our paper. This elaboration can be found in lines 102-104 on page 3. Furthermore, we have incorporated a mathematical description of the metrics, available for review in lines 546-550 on page 13.

Additionally, we concur entirely with the critique regarding the use of accuracy as a terminology. Given that accuracy is a distinct metric, conflating it with others could potentially confuse readers. We have therefore revised all related sections to address this concern. The amendments are located in line 330 on page 7; line 338 on page 8; line 446 on page 10; and lines 599, 603, and 614 on page 15.

Comments 3: Also, in the abovementioned reference, their review show:

[…Twitter spam detection is related to recall parameter with 23%, and F-measure comes next with 18%, precision with 17%, and accuracy 14%. …]

Hence, the section “Performance Comparison Based on Cost Changes” is a new viewpoint.

The authors can enhance this part of the discussion to reveal their innovation. 

Response 3:

Thank you for your response. The section you mentioned demonstrates how the system can be made scalable by adjusting costs based on changes. In situations where known spam is prevalent, we increase C(p, n), and conversely, we raise C(n, p) in opposite conditions to adjust the system. We have observed that increasing C(p, n) enhances the detection rate of known spam, while elevating C(n, p) improves the detection accuracy of unknown spam.

Although we have described these concepts in the text, it seems that the explanation was lacking in numerical detail. Reflecting on your comments, we have added a description of the numerical changes. These can be found in lines 653-658 and 660-662 on page 17.

Comments 4: The title “Scalable Learning Framework for Detecting New Types of Twitter Spam with Anomaly Detection and Decision Trees”

This title is a little strange. The whole article does not mention the scalability part. Moreover, Decision Trees is a method of Anomaly Detection and should not be written together.

Response 4:

Thank you very much for the comment that helped make the title more clear. Regarding the term 'scalable,' it relates to comment 3 previously addressed. Our system is equipped with a framework that allows flexible adaptation through cost adjustment. Details on this can be found in section 3.4, Scalable Spam Detection System. Additionally, explanations on experiments are provided in 4.3, Performance Comparison Based on Cost Changes.

As for the other comment concerning the decision tree, initially, decision trees are utilized for misuse detection, allowing their use alongside anomaly detection. However, the terms do not operate on the same level. Therefore, we plan to revise the title to incorporate decision trees under misuse detection. The revised title is as follows:

“Scalable Learning Framework for Detecting New Types of Twitter Spam with Misuse and Anomaly Detection”
